# Platelet Rich Plasma and Platelet-Related Products in the Treatment of Radiculopathy—A Systematic Review of the Literature

**DOI:** 10.3390/biomedicines10112813

**Published:** 2022-11-04

**Authors:** Eva Kubrova, Gabriel A. Martinez Alvarez, Yeng F. Her, Robert Pagan-Rosado, Wenchun Qu, Ryan S. D’Souza

**Affiliations:** 1Department of Physical Medicine and Rehabilitation, Mayo Clinic, Rochester, MN 55905, USA; 2Department of Anesthesiology and Perioperative Medicine, Division of Pain Medicine, Mayo Clinic, Rochester, MN 55905, USA; 3Department of Pain Medicine, Mayo Clinic, Jacksonville, FL 32224, USA

**Keywords:** platelet-rich-plasma, platelet lysate, plasma-rich-in-growth factors, radiculopathy, radicular pain, orthobiologics

## Abstract

Back pain with radicular symptoms is associated with detrimental physical and emotional functioning and economic burden. Conservative treatments including physical, pharmacologic and injection therapy may not provide clinically significant or long-standing relief. Regenerative medicine research including Platelet rich plasma (PRP), Platelet lysate (PL) or Plasma rich in growth factors (PRGF) continues to develop, however evidence appraisal for treatment of radicular pain remains lacking. Thus, we performed a systematic review to evaluate the effectiveness of epidural steroid injections containing PRP or related products to treat radicular pain. Embase, PubMed/MEDLINE, Cochrane Central Register of Controlled Trials (CENTRAL), and Google Scholar databases were queried. Twelve studies were included in qualitative analysis, consisting of three randomized controlled trials and nine observational studies. The primary outcome was pain intensity, and secondary outcomes included functional improvement, anatomical changes on advanced imaging, and adverse events. All studies identified improved pain intensity and functional outcomes after epidural injection of PRP, PRGF and/or PL. Similar or longer lasting pain relief was noted in the PRP cohort compared to the cohort receiving epidural steroid injections with effects lasting up to 12–24 months. The Grading of Recommendations Assessment, Development and Evaluation (GRADE) analysis revealed a very-low certainty of evidence due to risk of bias, indirectness, and imprecision.

## 1. Introduction

Low back pain (LBP) is a leading cause of disability in adults and is associated with a significant economic burden [1]. In most cases, LBP can be multifactorial with etiological factors related to spinal musculature, facet joints, spinal ligaments, vertebral periosteum, and central cord or spinal nerve roots [2]. Compression or inflammation of the exiting spinal nerve roots may lead to radicular pain symptoms, commonly in the setting of a disc herniation or spondylotic changes of the spine [2]. First-line treatment options for radicular pain include rest, physical therapy, and anti-inflammatory or neuropathic oral medications [2,3,4]. If conservative or pharmacological therapy do not provide alleviation, consideration of epidural steroid injections (ESIs), minimally invasive surgical decompression techniques [5], or open surgical decompression for refractory cases may be pursued [2].

ESIs are one of the most common interventional procedures used to treat radicular pain secondary to a herniated disc or spinal stenosis [2]. Although there are studies showing moderate short-term symptomatic improvement after ESI, studies focusing on long term follow-up have conflicting evidence regarding a long-lasting benefit [6]. Moreover, there are some rare, but potential risks associated with these ESIs, including infection, post-dural puncture headache, hematoma, hyperglycemia, adrenal suppression, myopathy, and nerve injury [7]. Given the adverse effect profile in addition to the mixed evidence on long-term efficacy from ESI for radicular pain [8], alternative non-operative treatments such as injectable orthobiologics have been under investigation [9]. Orthobiologics are substances that may be obtained from autologous blood and subsequently injected into a particular painful area to facilitate an anti-inflammatory milieu, recovery, and healing of muscle, tendon, and bone injuries. In many cases these substances serve as natural anti-inflammatories and pain modulators [10].

Platelet rich plasma (PRP) is a type of orthobiologic that induces healing by facilitating mobilization of inflammatory cells that secrete cytokines and growth factors in addition to fibroblast-like cells that play an important role in tissue regeneration and maturation [11]. Platelet-derived growth factors (PRGF) and Platelet lysate (PL) are platelet-derived products used for their potentially similar clinical benefits [12,13]. PRP injections have been utilized in treating tendinopathies and osteoarthritis, as well as adjunct treatment in surgeries, with limited studies demonstrating favorable results for treating facet joint arthropathy, degenerative disc disease and sacroiliac joint-related pain [14,15,16]. Although there is emerging evidence regarding the use of PRP to treat chronic musculoskeletal conditions, the literature is scarce when it comes to its utility in treating radicular pain.

The primary objective of this systematic review is to assess the efficacy of epidural injections containing PRP or platelet-derived products for the treatment of radicular pain. With our current knowledge of the physiologic properties of PRP, we hypothesized that PRP injected in the epidural space would be associated with reduced pain intensity, improved physical functioning, and favorable radiological findings (e.g., resolution of disc herniation, reduced nerve root impingement).

## 2. Methods

### 2.1. Literature Search Strategy

The study protocol was registered in the International prospective register of systematic reviews (PROSPERO CRD42022315447). The articles were broadly queried from multiple electronic databases, including Embase, PubMed/MEDLINE, Cochrane Central Register of Controlled Trials (CENTRAL), and Google Scholar. Broad MeSH terms and Boolean operators were selected, including terms and synonyms for radiculopathy, platelet rich plasma, plasma rich in growth factors, platelet lysate, and epidural injection. A medical librarian (E.K.V.) experienced in systematic review methods performed and verified this search strategy (See Appendix B). In addition, a manual literature search was performed to supplement the above strategy. 

### 2.2. Study Selection

Inclusion criteria included randomized controlled trials, observational studies, and case series or reports that reported pain-related outcomes in patients with radiculopathy who received an epidural injection of PRP or other platelet-derived products. Exclusion criteria included review articles, animal studies, and non-peer reviewed publications or abstracts. Two independent reviewers (E.K., G.A.M.A.) selected abstracts and full-text articles from the aforementioned search strategies. Disagreements were resolved by a third reviewer (R.S.D.).

### 2.3. Data Extraction and Outcomes of Interest

Two reviewers (E.K., G.A.M.A.) extracted all relevant data independently and any disagreement was resolved by a third reviewer (R.S.D.). The following data were extracted: study year, study country, study design, funding, number of subjects, product characterization, injection volume, subject age, injection guidance and site, length of follow-up, and study outcomes. The primary outcome of interest was change in pain intensity after epidural injection with PRP or platelet-derived product. Secondary outcomes were changes in physical functioning, radiological changes, and adverse events. 

### 2.4. Assessment of Risk of Bias

Risk of bias was independently assessed by two reviewers (G.A.M.A, Y.F.H.). For randomized controlled trials (RCT), we used the Cochrane Risk of Bias Assessment Guide in reference to a RCT that randomly selected participants to either receive epidural PRP or placebo/steroid injection. In reference to this target trial, biases were assessed in random sequence generation, allocation concealment, blinding of participants and personnel, blinding of outcome assessment, attrition bias due to missing data, reporting bias, and other biases. Each domain category was assigned a grade of low risk, high risk, or unclear risk.

We used the Newcastle–Ottawa quality assessment scale for observational studies with over ten patients [17,18]. Domains of the Newcastle–Ottawa Scale assess bias based on selection, comparability, and exposure. For length of follow-up, we determined that three months or more of follow-up would be adequate for outcome of interest to occur. For adequacy of follow-up of cohorts, we determined a follow-up of 75% or more of subjects would be adequate. A study can be awarded a maximum of one star for each numbered item within the Selection and Outcome categories. A maximum of two stars can be given for Comparability.

### 2.5. Assessment of Quality of Evidence

The GRADEpro software (Evidence Prime, Inc.; http://gradepro.org (accessed on 30 May 2022) was used to assess quality of evidence and certainty in findings for each outcome based on the Grading of Recommendations, Assessment, Development and Evaluations (GRADE) criteria. Level of evidence can be classified as high, moderate, low, or very low. RCTs are considered high-level of evidence. This can be downgraded because of deficiencies in the following domains: risk of bias, inconsistency, indirectness, imprecision, and publication bias. 

## 3. Results

### 3.1. Characteristics of Included Studies

A total of 12 studies fulfilled criteria for inclusion, which included three randomized controlled trials (RCT), four prospective observational studies, one non-randomized comparative study, two retrospective studies, and two case reports (Table 1). The study selection process is displayed in a PRISMA flowchart diagram (Figure 1).

A total of 1257 subjects were enrolled in all the included studies, out of which at least 1092 subjects received epidural injection with PRP or PRP-related products such as PL or PRGF. The remainder of patients received ESIs. A total of six studies with 218 patients received epidural PRP [4,21,22,23,24,26]. Lemper et al. used PRP diluted in platelet-poor plasma (PPP) in one patient [17]. Centeno et al. used PL in a combination with steroid (50% PL, 25% of 4% lidocaine, 25% 100–200 ng/cc hydrocortisone) in 470 patients [6] and Rawson et al. used PL alone but combined with injections of PRP to the facet joints and paraspinal muscles in two patients [18]. Three studies with a total of 401 patients received PRGF [19,20,25].

Patients’ age ranged between 16–91 years; however, average age in most studies was between 53.6–68 years. Time of follow-up ranged from six weeks to 24 months. One study included a subject with cervical and lumbar radiculopathy [17] and another two studies [20,25] enrolled subjects with either cervical or lumbar radiculopathy. The remaining studies included only lumbar radiculopathy [3,4,6,18,21,22,23,24,26].

### 3.2. Product Characterization

Three studies with 401 participants used PRGF [19,20,25], one study with 470 patients used PL [6] and one study used PRP and PL in two subjects [18]. Centeno et al. used PL that was prepared by one cycle of thaw-freeze method.

Some studies described in detail the sequence and methods of PRP or platelet-derived product [6,18,19,21,22,23,25], and one study referred to their previous published protocols on PRP preparation [26] while others did not provide any details [4,17,20]. In the studies using PRGF, products were activated with calcium chloride following centrifugation and before application [19,25]. The volumes of epidural injections ranged from 1.5 [25]–27 mL [17] in the cervical spine and 2 [19]–16.5 mL [23] in the lumbar spine (Appendix A).

### 3.3. Injection Procedure

Injections were fluoroscopy-guided in most studies [4,6,18,19,23,24,25,26], although two studies [22,26] used ultrasound guidance, and one study used computed tomography (CT) guidance [21]. The majority of studies injected subjects at one time point but five studies [17,18,20,24,25] performed repeat injections. Kirchner et al. reported that they performed up to five repeat injections in one patient [25].

All studies injected PRP products into the epidural space while five studies injected other structures in addition to epidural space such as surrounding paraspinal muscles [17], intervertebral discs and facet joints [19,25], and posterior spinal ligaments and facet joints [18]. Kirchner et al. also performed cervical and lumbar intraosseous vertebral injections in addition to epidural injections [25]. In a study by Machado et al., an epidural injection was performed in 76% of subjects, intradiscal injection in 22% of subjects, intramuscular in 100% of subjects, and facet injections in 98% of subjects [26]. 

### 3.4. Outcome Measure Tools

Six studies used the Visual analogue scale (VAS) to measure pain intensity [4,19,20,22,23,26] while one study used the Lattinen Index score [24]. The Lattinen test or index assesses chronic pain across five categories: pain intensity, pain frequency, use of analgesics, patient activity and hours of sleep. Each category is scored from 0 to 4, with a maximum score of 20 and a decrease in the score indicates an improvement in patient status [27]. Other pain intensity scales included Numeric Pain Score (NPS)/NRS [6,25]. COMI Pain Score (CPS), and OSWESTRY Pain Score (OPS) [25].

For functional outcomes, three studies used the Oswestry Disability Index (ODI) [21,22,25], one study used the Functional Rating Index (FRI) and Single assessment Numeric Evaluation (SANE) [6], and one study used MacNab criteria (Excellent, Good, Fair, Poor) [20]. Other tools to measure physical functioning outcomes were Modified Oswestry Disability Questionnaire (MODQ) [4,26], Physical Performance Test (PPTs) [22], Short-Form 36 (SF-36) with subscores Bodily Pain (BP) and Physical Functioning (PF) [22,23], Rolland Morris Disability Questionnaire (RMDQ), and COMI Disability Score (CDS) [25].

### 3.5. Primary Outcome—Change in Pain Intensity

Three RCTs [22,23,24] and one non-randomized comparative trial [21] showed that both ESI and epidural injection with PRP improved pain scores as measured by NRS [21] and VAS [22,23] and Lattinen index [24] at measured time points, with no significant differences between the two groups at six weeks and 12 months in two studies [21,22]. Benitez-Nunez et al. [24] reported that there was an initial pain relief in the Lattinen Index score response in the ESI group at one week that lasted one month (from mean 15.5 to 4.2 at one week and 3.0 at one month), but it was followed later by diminishing pain relief, while the PRP group experienced beneficial effects maintained at 12 months (mean score 15.6 to 12.3 vs. PRP score 15.9 to 1.7, *p* < 0.001). Similarly, Ruiz-Lopez et al. [23] reported that VAS improved in both groups, however, at one month the ESI group had mildly better improvement from mean 7.18 (±0.95) to 4.4 (±0.92) versus 7.48 (±1.12) to 5.2 (±0.69) in the PRP group, while PRP group had lower VAS scores at three and six months (6.28 ± 0.86 vs. 5.7 ± 0.97 and 7.53 ± 0.6 vs. 6.08 ± 0.99 in ESI vs. PRP, respectively). Average pain improvement ranged between 19 to 89% in the PRP cohort versus −5% to 66 % in the ESI groups at the last follow-up [22,23,24]. Length of follow-up ranged between six weeks to 12 months in the comparative studies. 

Overall pain scores also improved in all eight non-comparative observational studies [4,6,17,18,19,20,25,26], but one study reported a subset of ‘nonresponders’ which was about 15% of the studied patients [20]. Length of follow-up of non-comparative studies was between 1–24 months. The average percent of pain relief ranged from 32.7–100% or by at least 2 points in NRS at longest follow-up periods.

### 3.6. Secondary Outcome: Functional Outcomes

A total of eight studies reported improved physical functioning and disability outcomes [4,6,20,21,22,23,25,26] while the other studies did not assess functional outcomes. 

Functional score assessed by SF-36 was improved at final follow-up at 6 months and 12 months in two RCT studies [22,23]. In a study by Xu et al. [22] median PF subscores improved from 65 to 90 at 1 year in the steroid group vs. 60 to 90 in the PRP group. BP subscore improved from 41 to 74 at 1 year in both groups (*p* < 0.001). Ruiz-Lopez et al. [23] reported significantly better scores in all subgroups at 6 months in the PRP group compared to baseline and compared to the steroid treated subjects. Physical component summary mean score was 141.1 (±70.18) at baseline in the steroid group vs. 151.71 (±84.24) at 6 months (*p* = 0.39), and 140 (±75) at baseline vs. 226 (±61) at 6 months in the PRP group (*p* = 0.001; intergroup *p* = 0.0001).

Bise et al. [21] observed that ODI scores were similarly improved at six weeks without any significant differences between the steroid and PRP groups with the median percentage ODI decrease of −27 in the ESI cohort and −25 in the PRP cohort (*p* = 0.314). Xu et al. [22] reported median ODI improvement at 1 and 12 months but no significant differences between the two groups (median 27 [21–43] at baseline vs. 20 [17–40] at 1 year in the steroid group; 35 [26–44] vs. 19 [15.5–30] in PRP; *p* < 0.001) [22]. In addition, Xu et al. reported sustained improvement of the PPT score starting at one month and at a final follow-up at 12 months [22]. MacNab score improved at 12 months in one study from “poor” at baseline to “fair” at 2 months and “excellent” at 1 year in a non-comparative study [20].

Bhatia et al. reported MODQ index improved at three weeks and three months with median of 48 at baseline to 35 at 3 weeks and 28 at 3 months [4]. Another study showed that RMDQ improved at 12 weeks from 18.0 to 10.98 (*p* < 0.001), and this was sustained at 52 weeks [26]. Centeno et al. reported FRI scores improved at all time points from one month through 12 months. The change in FRI met the minimal clinically important difference at every time point except at one month and SANE improved by 49.7% at 24 months [6].

Overall, two [21,22] of three comparative studies assessing functional scores did not report any difference between functional outcomes among PRP product and ESI groups. One study reported significantly better outcomes in the physical function components of the SF-36 scale in the PRP group versus ESI (*p* < 0.001) [23].

Improvements in functional scores were observed as soon as three weeks [4], one month [6,22] and six weeks [21] post-injection and a total of five studies found that improvements were sustained at follow-up at 12 months [6,20,22,23,26] and 24 months [6]. Of these studies with long follow-up periods, three used PRP [22,23,26], one used PL [6] and one used PRGF [20].

### 3.7. Secondary Outcome: Anatomical Changes on Advanced Imaging

One study injecting PL into epidural space and PRP into facet and ligamentous structures reported improvements observed on advanced imaging studies [18]. MRI was used to assess intervertebral discs at baseline and at 1 and 3 months after second PL injections (2 and 5 months after first treatment) and found disc resorption in both subjects. Another study using epidural PRP injections did not report any differences in imaging studies in either PRP or ESI groups [24].

### 3.8. Secondary Outcome—Adverse Events

One retrospective study [6] showed a 6.3% rate of mild adverse events (AEs) which included post-procedural inflammation, soreness, muscle tightening, stiffness and/or numbness. Other AEs included positional headaches, lightheadedness, or skin reactions. Of the three patients who reported symptoms consistent with dural puncture, two resolved with conservative care and the third resolved after receiving an autologous blood patch [6]. Another study reported post-procedural pain in one to three days [18] that resolved with symptomatic treatment. One study [24] reported mild AEs were five times more frequent in the cohort that received ESIs versus the PRP cohort.

### 3.9. Risk of Bias Assessment

We used the Cochrane collaboration Risk of Bias tool to assess risk of bias in three RCTs [22,23,24] (Figure 2). Ruiz Lopez et al. had overall low risk of bias for all categories. Xu et al. had a high risk of bias in two categories which included blinding of participants and personnel and outcome assessment. Similarly, Benítez Núñez et al. had several concerns regarding high risk of bias in blinding of participants and personnel and outcome assessments, and unclear risk in other domains.

The Newcastle–Ottawa scale was used to evaluate non-randomized comparative and non-comparative observational studies [4,6,17,18,19,20,21,25] (Table 2). Only one study received a star for the comparability domain, indicating a high risk of bias for comparability among most observational studies [21].

Studies were evaluated based on the Newcastle Ottawa Quality assessment scale for cohort studies.

### 3.10. GRADE Quality of Evidence

We combined grading of outcomes for RCTs and non-randomized studies together. Given the generally low evidence level from non-randomized studies, along with risk of bias (issues with blinding, non-randomized studies), indirectness (wide range of ages for patients, many studies did not compare ESI and PRP injections), imprecision (mixed results for certain outcomes), a very low-quality GRADE certainty of evidence was assigned to all outcomes of interest (Table 3).

## 4. Discussion

This systematic review highlights improvement in pain scores following epidural injection with PRP, PRGF or PL in all included studies. In the short term (approximately 1 month follow-up), studies highlight that epidural PRP or platelet-derived product injections are associated with improvement in radicular pain similar to ESI. The proposed mechanisms behind PRP and platelet product-induced pain relief partially overlap with the effects from ESI injections. Based on current knowledge, both PRP and ESI are thought to act through locally diminishing levels of inflammatory factors such as phospholipase A2 (PLA2), interleukin-1α (IL-1α), IL-1β, IL-6, IL-8, tumour necrosis factor-α (TNF-α), and prostaglandin E2 (PGE2) [28,29]. These factors are typically elevated in the tissues surrounding bulging or herniated discs leading to nerve root irritation and radicular pain [30,31]. In addition, growth factors and cytokines released from platelet α granules with local anti-inflammatory, anti-apoptotic and analgesic effects (e.g., through cytokines such as IL-4 or IL-10) [19,25], structural changes such as herniated disc resorption [18], extracellular matrix production (ECM) and neural regeneration [23,32,33,34] are thought to be main mechanisms responsible for PRP benefits. 

Another finding from two comparative studies is that epidural PRP or platelet-derived products may provide longer lasting pain relief compared to ESI [23,24]. This is concordant with evidence in other musculoskeletal conditions (e.g., knee osteoarthritis), where studies highlight that PRP injection may achieve longer-lasting pain relief compared to steroids or hyaluronic acid [35]. The reasons for these findings are unclear, however, this could be due to the aforementioned mechanisms intrinsic to platelet products such as immunomodulation, facilitation of structural changes and tissue healing. Notably, in chronic stages, platelet cytokines may reinitiate inflammatory cascades that can lead to a reparative process [17]. In addition, PRP injected epidurally can exert paracrine activity on surrounding structures including intervertebral discs, facet joints, spinal ligaments, which can further contribute to pain relief [20,26]. Kirchner et al. proposed mechanisms of action of PRGF on discs and facet joints may include ECM recovery and anti-apoptotic effects through platelet-released cytokines (fibroblast growth factors (FGF), platelet-derived growth factor (PDGF), insulin-like growth factor-1 (IGF-1), vascular endothelial growth factor (VEGF), transforming growth factor (TGF-β), hepatocyte growth factor (HGF), connective tissue growth factor (CTGF), and nerve growth factor) and fibrin matrix [36].

Physical function and level of disability were improved in eight studies after epidural PRP injection. Further, one comparative study reported significantly better improvement in all physical function SF-36 subscores in the PRP cohort compared to the ESI cohort [23]. Additionally, one study noted improvements in the MRI findings and resorption of the extruded discs after two PL injections in both of their subjects [18]. It is well-known that disc components may require surgical retrieval in cases resistant to conservative treatments, but nearly complete disc herniation resolution may also occur spontaneously as a natural history of the disease in about 40 to over 70% of subjects within 1–2 years [28,37,38]. The mechanism of disc resorption is poorly understood, and possibly related to either dehydration of nucleus pulposus, retraction or complete separation away from the nerve root area [37]. Given that only two studies in our review performed radiological MRI evaluation after PRP injection, it is unclear if disc resorption was natural versus if there were any effects attributable to the PRP. However, disc resorption occurred in a relatively short period of time (2–5 months). 

Notable differences in cost-effectiveness and procedural time may be present. Preparation of PRP, PL and PRGF involves use of commercially available devices. PRP is a concentrate of platelets collected from a venous anti-coagulated blood sample of 40–60 mL, and subsequently centrifuged and processed to create a product that is on average 2–8 times more concentrated than normal platelet plasma concentration with small or large amount of white blood cells (leukocyte rich vs. leukocyte poor PRP) [39,40]. PRGF is a leukocyte-free PRP product that is obtained from the supernatant above the buffy coat after blood centrifugation and then activated with calcium chloride to promote platelet degranulation and release of growth factors [12]. PL is a growth-factor rich product isolated from peripheral blood but without the cell debris. It is created by isolating the supernatant after initial centrifugation, and then the platelets in the plasma are lysed by freezing them at −80 °C and then thawed prior to a second centrifugation. This supernatant is then filtered and used for injection [13]. These preparation protocols for orthobiologics may involve significant cost, labor, and procedural time, which may impact implementation of this therapy into standard clinical practice.

Several considerations warrant mention for PRP dosing. First, inconsistency was noted with PRP products between studies due to modifiable variables such as concentrating device used, volume of blood drawn, concentrations of PRP, use of activating agents, incorporation of white blood cell-rich or poor adjuncts. Heterogeneity was also noted with non-modifiable factors such as patient’s age, comorbidities, and individual properties of the plasma and PRP proteins. Studies in this review typically reported venous blood and PRP volumes, but only one study reported PRP analysis of concentrations of platelets (520,000/µL ± 114,250) and leukocytes (310,000/µL ± 293,000). The final products consisted of 2.7–25.3% of the volume of the originally collected venous blood. Common product to venous blood ratio to yield about 1,000,000 platelets/µL in PRP is with about 30 mL venous blood collected and about 2–5 mL injected (6–16% original volume). Typically, reported platelet concentrations thought to be effective are about 500,000–1,000,000 per µL [41,42]. Given the large variability in the volume injected, it would be important to know whether the dosing was sufficient in the included studies. 

In terms of injection frequency, it has been previously reported that in patients with acute radicular pain who do not have initial large benefit from epidural steroid injection (<three months), repeat injections may contribute to a cumulative response [43]. Some of the included studies also performed repeat epidural injections with PRP and platelet derived products, which generally revealed continuously improved outcomes over time, perhaps due to cumulative effects [20,24,25]. It is also plausible that disc herniation causing radiculopathy or radiculitis may improve over time without any intervention, which is consistent with its natural history [44].

In this review, epidural injection of PL/PRP/PPP/or PRGF was safe. The most common complaints were injection site pain, redness, swelling, stiffness, and soreness. There were three patients who had symptoms from dural puncture that resolved with conservative treatment or epidural blood patch [6]. This is comparable to the reported complications associated with epidural steroid injections [45,46,47]. Importantly, no episodes of arachnoiditis or other serious adverse events (neuraxial hematoma, infection, nerve injury) were reported in patients who received epidural injection of PL/PRP/PPP/PRGF, although a true safety assessment is hindered by the small sample size from included studies. Another major adverse event that warrants discussion is spinal cord infarction due to vaso-occlusive events. This is particularly important because spinal cord infarction due to vaso-occlusive events from particulate steroid suspension in transforaminal epidural injections is a well-documented and a devastating complication [47]. Since the primary function of platelet is clot formation, we recommend that clinicians do not administer epidural PRP/PRGF/PPP using a transforaminal approach. Centeno et al. proposed that because the processing of PL involves filtration of platelets and other cellular products, epidural PL may not carry the potential for platelet aggregation and, hence, may carry a decreased risk of vascular occlusion and spinal cord infarction compared to other platelet-derived products. Although PL does not contain any platelet membranes, the presence of some of the membrane proteins may likely still affect platelet behavior in vivo. It has been observed anecdotally that PL can form aggregates in vitro and may require further filtering [48]. Further, one study reports higher risk of clotting with PL versus PRP if no fibrinogen depletion is performed [49]. Therefore, caution is also advised with epidural injection of PL due to potential risk for spinal cord infarction. 

## 5. Limitations

There are several limitations. First, the included studies contained small sample sizes. There was overall high risk of bias and lack of comparator arms in most studies. Second, there was a substantial clinical and methodological heterogeneity between and within the studies in terms of participant selection, PRP preparations (different preparation strategy), volume of peripheral blood used for product preparation, total volume, type of product (PL/PRP/PPP/PRGF) or their combinations, structures injected (cervical/lumbar epidural spaces, trigger points, intervertebral disc, facet joint, or posterior spinal ligament, intraosseous injections) or their combinations, mode of image guidance (ultrasound/Fluoroscopy/CT), and follow-up periods. Third, industry funding may add additional bias [6].

## 6. Future Considerations

Larger studies devoted to safety and adverse effect assessment are warranted, particularly with injection into epidural space. The authors query the possibility of arachnoiditis in the event of inadvertent intrathecal administration of orthobiologics, although future studies are needed to evaluate for this possible complication. Future well-powered RCTs are warranted to confirm if PRP injection into the epidural space offers longer duration of relief compared to ESIs. Dose–response studies are important to identify the optimal concentration and volume of orthobiologics for epidural administration. Detailed characterization of the products should be included in all studies. In addition, a head-to-head comparison of PRP, PL and PRGF would help clarify the most efficacious product for this particular condition. Further, comparison of technique of epidural injection (interlaminar versus transforaminal) of platelet-derived products should also be performed. Future studies should maintain consistency in selected patient populations, such as acute versus chronic pain, and cervical versus lumbar radiculopathy. Inclusion of advanced imaging findings such as MRI would help with assessment of anatomical changes and safety monitoring. 

## 7. Conclusions

Overall, in patients with radicular pain, epidural injection with platelet-derived products may decrease pain intensity and improve functionality similar to epidural steroid injections. Anatomical resolution of pathology causing radicular symptoms is possible, but was only noted in one study and would require further studies for adequate evaluation. Based on comparative studies, the initial response of ESI and platelet-derived products is similar, however, the duration of relief from epidural injections with platelet-derived products may be longer compared to ESI, although this finding was inconsistent across comparative studies. Studies also suggest that epidural injection with PRP, PL or PRGF may be safe with only mild adverse events, although the current analysis consisting of only 1092 patients receiving the treatment arm is insufficiently powered to definitively make this conclusion. Given the nature and location of the treatment, vascular occlusion or arachnoiditis should be considered as a potential side effect and future studies with an adequate sample size are warranted to further assess their safety. Lastly, GRADE Quality of Evidence assessment showed that primary and secondary outcomes were assigned very-low GRADE certainty of evidence due to risk of bias, indirectness, and imprecision.

## Figures and Tables

**Figure 1 biomedicines-10-02813-f001:**
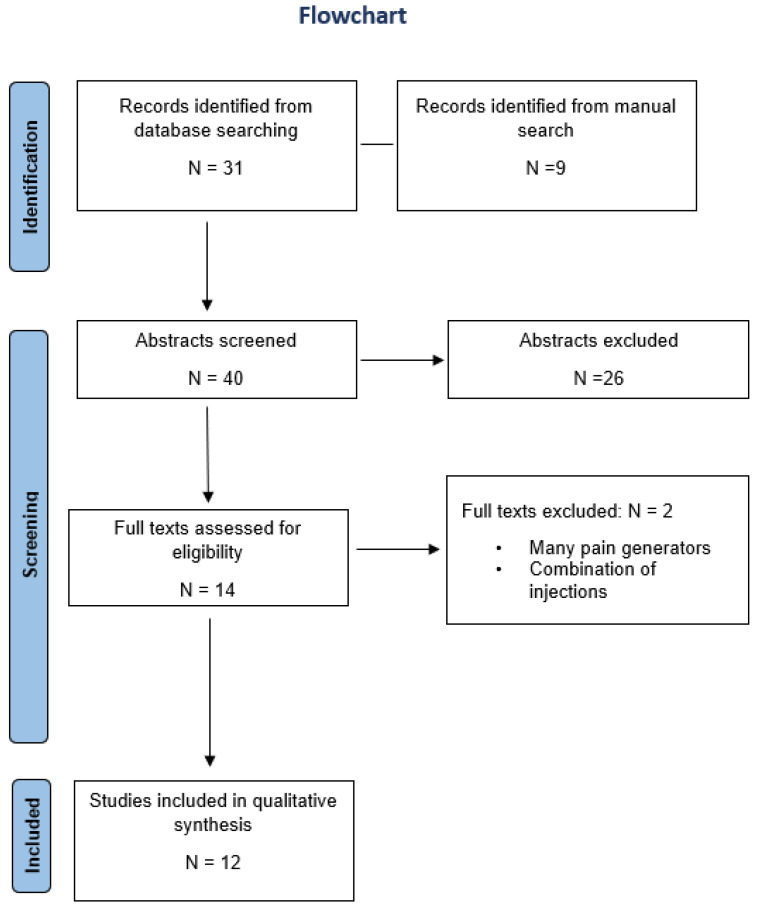
PRISMA 2009 flow diagram.

**Figure 2 biomedicines-10-02813-f002:**
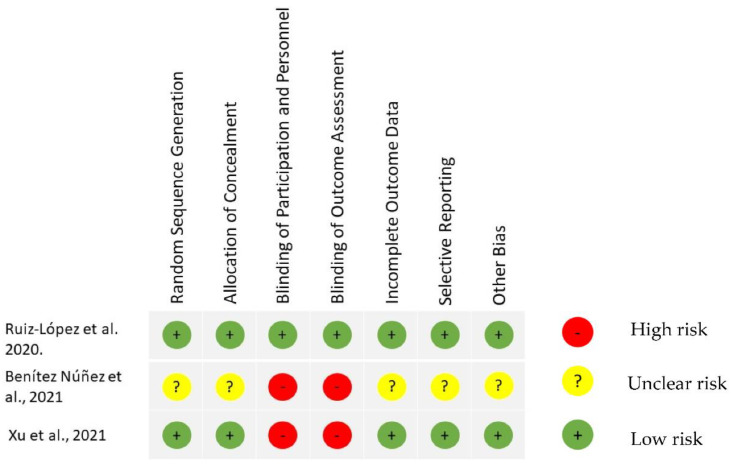
Risk of bias summary of the RCT-studies based on authors’ judgements of each item [22,23,24].

**Table 1 biomedicines-10-02813-t001:** Summary of findings of the included studies.

Author, Year	Country	Study Design	Study Funding	N of Subjects	Product and Volume Injected, Site	Subject Age	Injection Guidance and Site	Follow-Up	Outcomes
Lemper et al., 2012 [17]	USA	Case report	None reported	1	9 mL of PRP with 18 mL of PPP into cervical epidural space; trigger point and lumbar epidural injection volume not specified	35-year-old female, 10 W pregnant	Bilaterally C4-C6 epidural injections + in surrounding trigger points. Lumbar epidural injection 1 M later (level not specified).	3 M	1st injection: Cervical: pain improved by 50% next day and further at 1 W, reported better relief compared to prior ESIs 2nd injection: Lumbar: Pain completely resolved at 3 M. No AEs, delivered healthy child
Bhatia et al., 2016 [4]	India	Prospective uncontrolled study	None reported	10	5 mL of PRP	All subjects < 65 Y	Fluoroscopy-guided interlaminar lumbar epidural injection	3 M	Gradual improvement of VAS scores (avg 39.3% at 3 M), MODQ index (to <30%) and SLRT (<70) at 3 W and 3 M in all subjects. No AEs reported.
Kirchner et al., 2016 [19]	Spain	Prospective uncontrolled study	No funding. COI: Dr. Anitua is scientific director at BTI, who developed PRGF-Endoret technology	86	PRGF 4 mL into disc, 2 mL peri-durally, other into facet joint	47 females (median 58) 39 males (median 55)	4 mL intradiscal 0.5 mL intra-articular facet 2 mL transforaminal epidural lumbar injection of PRGF, fluoroscopic guidance	6 M	At 6 M 90.7% of patients showed an excellent VAS score (0–3), 8.1% showed a moderate improvement (3.1–6.5), 1.2% no improvement (6.6–10).
Centeno et al., 2017 [6]	USA	Retrospective study	Funded by Regenexx, LLC and the Centeno-Schultz Clinic.	470	PL: Injection of 3–5 mL of PL (50% PL, 25% of 4% lidocaine, 25%100–200 ng/mL hydrocortisone)	Mean 53.6 (SD 13.5)	Fluoroscopy-guided transforaminal or interlaminar lumbar injection	24 M	Subjects treated with PL reported significantly lower NPS (by AVG of 51% at 24 M) and FRI change scores at all time points 1 M–24 M (on AVG by 39.7%). FRI improvement met MCID at every time point except at 1 M. SANE improved in 72.7–77.1% subjects across time points. Total of 6.3% of subjects had mild AEs related to the procedure, no SAEs.
Correa et al., 2019 [20]	Colombia	Prospective uncontrolled study	Not reported	250	PRGF 10 mL in cervical or 12 mL in lumbar spine	Range 18–70	Epidural cervical level 30% (C6-C7) or lumbar level 70% (L4-L5 or L5-S1) Repeat injections at 6–8 W.	12 M	Mean VAS improved in 85% of patients (from AVG 8/10 to 2/10) at 12 M, MACNAB criteria improved from poor at baseline to excellent at 12 M. Opioid rescue use decreased from 96% at baseline to 0% at 12 M. 15% did not have improvement but no symptom worsening. No AEs reported.
Rawson et al., 2019 [18]	USA	Case report	No funding	2	1 mL of PRP to posterior spinal ligament and facet joints per structure 3 mL of PL into epidural space	S1: 31 S2: 38	Both subjects: PRP in posterior spinal ligaments and facet joints Subject 1: L4-5 interlaminar epidural injection of PL and S1 repeat injection Subject 2: L4-5 interlaminar epidural injection under fluoroscopic guidance x2	6 M	S1: Post-procedural pain for 3 D. At 4 W 50% pain relief. Repeat injection at S1 level at 4 W. Since 2nd injection complete symptoms resolution maintained at 6 M. MRI revealed incomplete resorption of disk herniation. S2: 1–2 D of postprocedure pain. 3 M after injection complete resolution of pain and function. No reported AEs. Repeated lumbar MRI showed almost complete resorption of disk material, with no evidence of ongoing neural impingement
Bise et al., 2020 [21]	France	Non-randomized comparative study	No funding	60 (30 ESI, 30 PRP)	2.5 mL of PRP	ESI—50 (SD 16) PRP 59 (SD 15)	CT-guided interlaminar lumbar epidural injections	6 W	A statistically significant improvement found in both groups (decrease by 35%) at 6 W. ODI median decreased by 25%. No significant difference was observed in the decrease in NRS and ODI scores between the two groups at 6 W. No SAEs.
Xu et al., 2021 [22]	China	RCT	No funding	124 (61 in PRP group, 63 in ESI group)	PRP: 3 mL of PRP Steroid: 3 mL steroid + anesthetic	PRP: 56 (44.5–60) Steroid: 56 (50–59)	Ultrasound-guided lumbar transforaminal epidural injection	12 M	Statistically significant improvement in VAS, PPT, ODI and SF-36 at 1 M and maintained at 12 M. No significant intergroup differences in outcomes. No AEs reported.
Ruiz-Lopez et al., 2021 [23]	Spain, Taiwan	RCT, double blind	No funding	50 (25 in triamcinolone group, 25 in PRP group)	LR-PRP: LR-PRP 16.5 mL + 3 mL contrast Steroid: 20 mL of triamcinolone (60 mg) + 3.5 mL contrast	LR-PRP group 68 (13.06) Steroid group 61 (12.6)	Fluoroscopy-guided caudal epidural injection S3-S4 level	6 M	There was a significant reduction in the VAS score in both groups. VAS score was lower at 1 M in the ESI group, but scores were lower in the LR-PRP group at 3 M and 6 M. SF-36 at 6 M showed significant improvement in the LR-PRP group. No AEs reported.
Benítez Núñez et al., 2021 [24]	Cuba	RCT	No funding	93 (46 in PRP group, 47 in ESI group)	5 mL of autologous ozonized PRP or 5 mL mixture including bupivacaine and 40 mg of triamcinolone	85% of subjects were between 18–45, 15% were older than 45	Fluoroscopy-guided lumbar interlaminar epidural injection. Repeat injections after 1 W.	12 M	There was a gradual improvement in the Lattinen Index score for the PRP group, maintained at 12 M. In the ESI group, there was rapid improvement in Lattinen Index score at 1 w-1 M with subsequent worsening at 3–12 M. No radiological improvement in either group. Mild AEs were 5× higher in the ESI group.
Kirchner et al., 2021 [25]	Spain	Retrospective Observational Study	No funding. COI: Dr. Anitua is scientific director at BTI, who developed PRGF-Endoret technology	65 (18 with cervical and 47 with lumbar back pain)	Cervical epidural: 1.5 mL Lumbar epidural: 2 mL	Cervical pain: mean 54 Lumbar pain: mean 51	Cervical injections: 1. Intraosseous (11% pts) 2. Intradiscal (100% pts, 2–4 discs) 3. Facet joint (44% pts) 4. Epidural (80% pts) Lumbar injections: 1. Intraosseous (9.4% pts) 2. Intradiscal (100% pts, 1–4 discs) 3. Facet joint (74%) 4. Epidural (83% pts) All pts received at least 2 injections (1 M apart) Fluoroscopy-guided	1–24 M	Statistically significant Improvements in all 9 scores (without stratification) and 99% subjects had symptoms improvement after stratification. MCID was achieved in all subgroups for NRS (2 points decrease) and ODI. No SAEs.
Machado et al., 2021 [26]	Brazil	Prospective, Uncontrolled Study	No funding	46	2 mL of PRP for foraminal injections, 5 mL for caudal epidural injection, (2 mL for each facet joint, 2 mL for each site of paravertebral muscles and 1 mL for intradiscal injection)	55.1 (SD 15.2)	Injection sites determined by exam and imaging. Fluoroscopy-guided transforaminal or caudal epidural (76%) and intradiscal (22%) injection. Fluoroscopy and US for facet (98%) and IM (100%) injections.	12 M	VAS improved at 2 W and RMDQ at 12 W and both were sustained at 52 W. Mean VAS decreased by 35% and RMDQ by 40% at 52 W. Pain medication use decreased at 52 W. Opioid medication use was significantly decreased and number of subjects taking them decreased from 35 to 12 after the procedure. No AEs observed. Total of 3 subjects underwent spine surgery in 1 Y.

AE—adverse event; AVG—average; BP—bodily pain; COI: conflict of interest; D—day(s); ESI—epidural steroid injection; FRI—Functional Rating Index; IM—intramuscular; LR-PRP—leukocyte rich platelet rich plasma; MCID—minimal clinically important differences; MODQ—modified Oswestry Disability Questionnaire; M—month(s); mL—milliliter; NPS—Numeric pain score; NRS—Numerical rating scale; ODI—Oswestry disability index; PL—platelet lysate; PPP: platelet poor plasma; PRP platelet rich plasma; PPT—physical performance test; Pts—participants; RCT—randomized controlled trial; RMDQ—Roland Morris Disability Questionnaire; SANE -Single Assessment Numeric Evaluation; SAE—serious adverse event; SF-36—36-item short form health survey; SLRT—straight leg raise test; VAS—Visual analogue scale; W—week(s); Y—year(s).

**Table 2 biomedicines-10-02813-t002:** Quality rating for the non-RCT studies.

Author	Year	Selection	Comparability	Exposure/Outcome
Bhatia et al.	2016	**	-	**
Correa et al.	2019	**	-	*
Bise et al.	2020	***	*	*
Centeno et al.	2017	**	-	*
Machado et al.	2022	**	-	**
Kirchner et al.	2016	**	-	**
Kirchner et al.	2021	**	-	**

Notes: * Each * indicates that the study fulfills a criterium on the Newcastle-Ottawa Scale. A study can be awarded a maximum of one star for each numbered item within the Selection and Outcome categories. A maximum of two stars can be given for Comparability. In total, a maximum of four stars can be given for the Selection domain, maximum of two stars can be given for the Comparability domain, and maximum of three stars can be given to the exposure/outcome domain.

**Table 3 biomedicines-10-02813-t003:** GRADE assessment.

Certainty Assessment							№ of Patients		Certainty	Importance
№ of Studies	Study Design	Risk of Bias	Inconsistency	Indirectness	Imprecision	Other Considerations	PRP or PL or PRGF/Total	Steroids/Total		
Pain (Follow-up: Range 6 Weeks to 12 Months)										
12	RCTs and non-randomized controlled trials	Serious ^a^	Serious ^b,c^	Serious ^b,c^	Serious ^d^	None	1092/1257 (86.7%)	165/1257	⨁◯◯◯ Very low	IMPORTANT
Functional scores										
12	RCTs and non-randomized controlled trials	Serious ^a^	Serious ^b,c^	Serious ^b,c^	Serious ^d^	None	1092/1257 (86.7%)	165/1257	⨁◯◯◯ Very low	IMPORTANT
Safety										
12	RCTs and non-randomized controlled trials	Serious ^a^	Serious ^b,c^	Serious ^b,c^	Serious ^d^	None	1092/1257 (86.7%)	165/1257	⨁◯◯◯ Very low	CRITICAL

CI: confidence interval. Explanations. a. Most observational studies introduced a high risk of bias in “selection,” “comparability of cohorts,” and “assessment of outcome” and 2 RCTs had high risk of bias in blinding of participants, personnel, and outcomes. b. Some studies performed epidural injections only, but other studies performed additional injections at other sites such as trigger points/intramuscular injections, facet joints, intervertebral discs, or intraosseous injections. c. In some studies only epidural PRP or PRP related product were injected. In other studies combination of treatments was used such as PL/PRP with a local anesthetic and steroids, PRP diluted in PPP. d. Wide range and overlap in confidence intervals.; “⨁◯◯◯” reports on level of certainty. Here the “⨁” means one positive point was given out of 4 total possible.

## Data Availability

No new data were created in this study. Data sharing is not applicable to this article.

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
