# Peer review of "Platelet Rich Plasma and Platelet-Related Products in the Treatment of Radiculopathy—A Systematic Review of the Literature"

_biomedicines, 2022, doi:10.3390/biomedicines10112813_

Round 1

Reviewer 1 Report

The aim of this systematic review was to analyze the role of platelet rich plasma (PLP) and platelet-related products in the treatment of radiculopathy in comparison to epidural steroid injection of. Therefore, the authors included 12 studies (page 3, chapter: results) in there review, but in the flowchart (figure 1) they demonstrated 11 included studies. This confused the readers. 

The reports in this difficult topic are missing. The authors present there review very clearly and report is objective, but their findings are not proved the conclusion, that the epidural injection of PRP and platelet-related products are safe. The authors may be more careful with this art of state. 

Nevertheless, the authors present an interesting point of view for treatment radiculopathy. The article is well written und well discussed. So, this manuscript has some rights to be recognized to be published.

Reviewer 2 Report

The importance of the work is clear, and it seems interesting to the community. Some minor issues should be addressed by the authors as listed below:

- PL and PRGF must be defined in the introduction

- Conclusions should be improved

- Give some suggestions for further studies
